# Haplotype Structures and Protein Levels of TGFB1 in HPV Infection and Cervical Lesion: A Case-Control Study

**DOI:** 10.3390/cells12010084

**Published:** 2022-12-25

**Authors:** Kleber Paiva Trugilo, Guilherme Cesar Martelossi Cebinelli, Érica Romão Pereira, Nádia Calvo Martins Okuyama, Fernando Cezar-dos-Santos, Eliza Pizarro Castilha, Tamires Flauzino, Valéria Bumiller-Bini Hoch, Maria Angelica Ehara Watanabe, Roberta Losi Guembarovski, Karen Brajão de Oliveira

**Affiliations:** 1Laboratory of Molecular Genetics and Immunology, Department of Pathological Sciences, Biological Sciences Center, State University of Londrina, Londrina 86057-970, PR, Brazil; 2Laboratory of Mutagenesis and Oncogenetics, Department of Biological Sciences, State University of Londrina, Londrina 86057-970, PR, Brazil; 3Laboratory of Research in Applied Immunology, State University of Londrina, Londrina 86057-970, PR, Brazil; 4Laboratory of Human Molecular Genetics, Department of Genetics, Federal University of Paraná, Curitiba 80060-000, PR, Brazil; 5Laboratory of Studies and Applications of DNA Polymorphisms and Immunology, Department of Pathological Sciences, Biological Sciences Center, State University of Londrina, Londrina 86057-970, PR, Brazil

**Keywords:** polymorphism, rs1800468, rs1800469, rs1800470, rs1800471

## Abstract

This study aimed to verify the role of *TGFB1* variants (c.–1638G>A, c.–1347C>T, c.29C>T, and c.74G>C) in HPV infection susceptibility and cervical lesions development, and their impact on TGFB1 cervical and plasma levels. *TGFB1* genotypes were assessed with PCR-RFLP and haplotypes were inferred for 190 HPV-uninfected and 161 HPV-infected women. TGFB1 levels were determined with immunofluorimetric assay. Case-control analyses were performed with logistic regression adjusted for possible confounders. Women carrying –1347TT or –1347CT+TT as well as those with 29CT, 29CC, or 29CT+CC were more likely to have HPV than –1347CC and 29TT carriers, respectively. Regarding haplotypes, the most frequent were *4 (GCTG) and *3 (GTCG). Women *4/*4 were less likely to have HPV than those with no *4 copy. Comparing the inheritance of *3 and *4, carriers of *3/*4 or *3/*3 were more susceptible to HPV than *4/*4. The TGFB1 plasma and cervical levels were higher in the infected patients. Plasma levels were also higher in infected women with low-grade lesions. HPV-infected patients carrying *3/Other and *3/Other+*3/*3 presented lower TGFB1 plasma levels than those with no copy of *3. TGFB1 variants could contribute to the comprehension of the TGFB1 role in HPV-caused cervical disease.

## 1. Introduction

Human papillomaviruses (HPVs) are small, non-enveloped double-stranded DNA viruses with an icosahedral capsid capable of infecting skin and mucosa epithelial cells, belonging to the Papillomaviridae family [1]. Commonly, HPV is the most common cause of sexually transmitted infections worldwide [2].

More than 200 HPV types have been identified and are classified into low-risk (LR) and high-risk (HR) HPVs depending on their oncogenic ability [3,4]. Persistent infection with high-risk HPVs is associated with several human carcinomas and considered the main cause of cervical cancer [1,5], which is the fourth most commonly diagnosed cancer, as well as the fourth leading cause of cancer death in women [6].

HPV-driven cancer is a small probability event because most infections are transient and can be cleared spontaneously by the host immune system [7]. Therefore, the local immune response is an important determinant of progression and disease outcome [8]. Cytokines play a crucial role in mounting and maintaining immune responses against a host of pathogens, including viral infections and tumors [9].

In this context, we highlight the transforming growth factor-beta 1 (TGFB1), a pleiotropic cytokine that plays an important role in several biological processes, including cell replication, differentiation, apoptosis, angiogenesis, and immune system regulation [10,11]. Its signaling pathway has also been established as essential for cancer progression because of its prominent role in the regulation of cell growth, differentiation, and migration [12].

The *TGFB1* gene is located in the 19q13.2 chromosomal region, comprises seven exons separated by six very large introns [13], and presents various sequence variations that can be classified as functional, non-functional, or with undetermined function. Until now, eight single nucleotide variations (SNVs) and one deletion/insertion variant have been reported to be associated with a functional impact on TGFB1 production [14]. Among them we highlight rs1800468 (c.–1638G>A, G–800A, g.4245G>A) and rs1800469 (c.–1347C>T, C–509T, g.4536C>T), both located on the *TGFB1* promoter region, and rs1800470, on codon 10 (c.29C>T, Pro10Leu, g.5911C>T), and rs1800471, on codon 25 (c.74G>C, Arg25Pro, g.5956G>C), both on the signal peptide sequence.

Genetic variations may alter gene expression, messenger RNA (mRNA) stability, alternative splicing, microRNA target sequence, protein exportation to the endoplasmic reticulum via signal peptides, or alter protein function when an amino acid is changed [15].

Much has been discovered about the role of TGFB1 in HPV infection, such as the immunosuppression caused by TGFB1 favoring infection or the cytokine increasing by the action of viral oncoproteins. However, many pieces are lacking for complete elucidation of the mechanisms involving TGFB1 participation in the infection, intraepithelial lesion development, and cervical cancer establishment. Therefore, looking for one of these pieces, this work analyzes four genetic variations of *TGFB1* (c.–1638G>A, c.–1347C>T, c.29C>T, c.74G>C) and their haplotype structures in HPV-uninfected and infected patients, and in patients who developed or did not suffer premalignant lesions caused by HPV. This study also seeks to verify the impact of the haplotype structures on plasma and cervical TGFB1 levels.

## 2. Materials and Methods

### 2.1. Patients and Samples

Participants in the current study were women (n = 351) who underwent outpatient cytology testing between 2013 and 2015 at an ambulatory colposcopy facility of the Intermunicipal Consortium of Health of the Middle Paranapanema, at the University Hospital and Clinic Center of the State University of Londrina, and Basic Healthcare Units in Londrina-PR, Brazil. Cytobrushes were stored in 2 mL of TE buffer (10 mM Tris–HCl, 1 mM ethylenediamine tetra-acetic acid (EDTA) pH 8.0). Peripheral blood was drawn into sterile syringes containing EDTA. All samples were kept at 4 °C. Individuals were divided into groups based on the cervical cytology-determined lesion grade and the presence or absence of HPV DNA. Each participant completed a structured questionnaire and signed an informed consent form.

### 2.2. DNA Extraction

Peripheral blood, 200 μL, was intended for DNA extraction and the remainder had the plasma separated and stored at −20 °C until TGFB1 dosage. Suspension of cervical cells in TE buffer was centrifuged and used for DNA extraction, while the supernatant was stored at −20 °C until TGFB1 dosage.

Genomic DNA was obtained from cervical cytobrushes using DNAzol (Invitrogen^TM^, Carlsbad, CA, USA) according to the manufacturer’s instructions, and stored at −20 °C. DNA was also extracted from peripheral blood using a Biopur Mini Spin Plus Kit (Biometrix, Curitiba, PR, Brazil). DNA concentration was measured on a NanoDrop 2000c spectrophotometer (Thermo Fisher Scientific, Waltham, MA, USA), and purity was assessed by the A260/A280 ratio.

### 2.3. HPV Detection by PCR

HPV was detected using MY09 (5′-CGTCCMAARGGAWACTGATC-3′) and MY11 (5′-GCMCAGGGWCATAA-YAATGG-3′) primers. They were designed to amplify a conserved region (450 base-pairs (bp)) in the HPV L1 gene region (GenBank Accession number: AJ236888) [16]. Human b-globin (268 bp fragment) was co-amplified using GH20 (5′-GAAGAGCCAAGGACAGGTAC-3′) and PC04 (5′-CAACTTCATCCACGTTCACC-3′) primers [17]. DNA-free reactions and HeLa cells (HPV18 integrated DNA) were used as negative and positive controls, respectively.

All PCR products were electrophoresed on 10% polyacrylamide and stained with silver nitrate (Appendix A).

### 2.4. Cervical Cytology

Based on the Bethesda System (2014), cytologic samples were graded at the Public Health System Laboratory. Patient samples were classified as low-grade squamous intraepithelial lesions (LSIL) or high-grade squamous intraepithelial lesions (HSIL). Otherwise, a designation of no lesions (NL) was given if cytology samples were normal, i.e., not indicated as having low- or high-grade squamous intraepithelial lesions, cervical carcinoma, atypical squamous cells of undetermined significance, or other atypical squamous cells that could not be excluded as high-grade squamous intraepithelial lesions [17].

### 2.5. TGFB1 Genetic Variants Genotyping

Following the procedure published by Jin et al. (2004) with changes [18], genetic polymorphisms were examined by PCR and then by restriction fragment length polymorphism (RFLP) analysis. InvitrogenTM (Carlsbad, CA, USA) provided all of the PCR supplies, and New England Biolabs^®^ (Ipswich, MA, USA) provided all of the restriction enzymes. Briefly, two primer pairs were created using the TGFB1 gene reference sequence (NCBI gene bank accession number NG 013364.1), one of which included the two promoter region variants (c.-1638G>A and c.-1347C>T) and the other of which included the signal peptide variants (c.29C>T and c.74G>C) [18]. The PCR conditions were the same for the two reactions. In all, 25 L of PCR buffer (1×), dNTP (0.1 mM), primers (0.2 mM), MgCl2 (1.0 mM), Taq DNA polymerase (1 U/reaction), and genomic DNA (about 3 ng/L) were used for both reactions. To assess for exogenous DNA contamination, PCR reactions were conducted along a negative control with no DNA addition. The promoter region polymorphisms were flanked by primers with the sequences 5′-GCAGTTGGCGAGAACAGTTG-3′ and 5′-CCAGAACGGAAGGAGAGTCAG-3′, generating an amplicon of 597 base pairs (bp) at 59 °C. For the enzymatic digestion of the c.-1638G>A polymorphisms, the restriction enzyme *HpyCH4IV* was used, yielding 402 and 195 bp fragments for the G allele (Appendix A), and the *Bsu36I* restriction enzyme was used, yielding 488 and 109 bp fragments for the C allele, for the genotyping of the c.-1347C>T polymorphisms (Appendix A). The manufacturer’s method was followed for the restriction conditions. The signal peptide region was targeted by the primers 5′-TTCCCTCGAGGCCCTCCTA-3′ and 5′-GCCGCAGCTTGGACAGGATC-3′. The annealing temperature was chosen at 62 °C. The 294 bp amplicon was cleaved into 161, 67, 40, and 26 bp fragments for the T allele and 149, 67, 40, 26, and 12 bp fragments for the C allele by the *MspA1I* restriction enzyme (Appendix A). The *BglI* restriction enzyme cleaved this same amplicon in 131, 103, and 60 bp fragments for the G allele from c.74G>C polymorphism and in 163 and 131 bp fragments for the C allele (Appendix A). The manufacturer’s recommendations were followed for restriction conditions. Amplicons and restriction fragments were examined by electrophoresis on silver-stained 10% polyacrylamide gel.

To verify the precision of the genotyping procedure, one person per genotype across all genetic variations examined was sequenced in a 3500 Genetic Analyzer^®^ (Applied Biosystems) using the BigDye^®^ Terminator v3.1 Cycle Sequencing kit (Applied Biosystems, Austin, TX, USA) and PCR-RFLP analysis on at least 5% of the entire sample was repeated, yielding 100% agreement between the results.

### 2.6. Haplotype Analysis

Using PHASE software version 2.1.1, recombination sites between TGFB1 SNV alleles were inferred [19,20]. Haploview version 4.2 was used to investigate linkage disequilibrium [21]. In MEGA7 software, the haplotype tree was created using the maximum parsimony approach [22].

### 2.7. TGFB1 Levels

Blood samples were centrifuged at 3000 rpm for 15 min and plasma was stored at −20 °C. Cervical mucus samples collected with cytobrushes were suspended in TE buffer, centrifuged at 3000 rpm for 15 min, and the supernatants were stored at −20 °C.

TGFB1 levels were determined using microspheres immunofluorimetric assay (Novex™, Life Technologies, Frederick, MD, USA) for the Luminex platform (MAGPIX™, Luminex Corp., Austin, TX, USA), according to the manufacturer’s instructions. Results were expressed as pg/mg of total protein, and cervical TGFB1 levels were normalized by the total protein in the supernatant. The plasma concentrations were reported as pg/mL.

### 2.8. Statistical Analysis

Pearson’s chi-square (χ2) test or Fisher’s exact test was used to assess differences in the frequency distributions of clinical and sociodemographic categorical data as well as *TGFB1* variants (alleles, genotypes, and haplotypes) between case groups and controls regarding HPV status and SIL diagnosis. To avoid type error I from multiple comparisons, Bonferroni’s correction was applied as a post-hoc test. The χ2 test was also used to evaluate Hardy–Weinberg equilibrium. Differences between groups for continuous variables were assessed by the Mann–Whitney test or Kruskal–Wallis test with Dunn–Bonferroni’s post-hoc test. Categorical data were expressed as absolute number (n) and percentage (%) and continuous data as median and interquartile range (IQR). Binary or multinomial logistic regression were carried out to independent association prediction between SNVs as explanatory variables and HPV status or SIL diagnosis as outcome variables. Possible confounders using the forced entry method adjusted logistic regression results. Confounders were chosen by the automated backward method with the elimination criterion based on the likelihood ratio statistic. Adjusted odds ratios (ORs) and 95% confidence intervals (CI) were estimated. All tests were two-tailed, with a *p*-value (*p*) < 0.05 as statistically significant. Statistical analyses were carried out using SPSS Statistics 25.0 software (SPSS, Inc., Chicago, IL, USA).

## 3. Results

### 3.1. Participant Characterization According to HPV Infection, Sociodemographic, and Clinical Data

First, 351 women were included in the study and categorized as HPV-infected (45.9%) and HPV-uninfected (HPV control group—54.1%). The HPV-infected women were divided into three groups based on cytological abnormalities detected and classified according to the Bethesda System classification as follows: no cervical lesion (NL, lesion control group—51.3%), LSIL (14.7%), and HSIL (34.0%) groups. For lesion analyses, five patients were excluded, one woman without a cervical cytology result and four women diagnosed with cervical cancer (Figure 1).

Patients’ features such as age, age at menarche, age at first sexual intercourse, pregnancies, oral contraceptive usage, marital status, sexual partners during the lifetime, and smoking status are summarized in Table 1. In this, a significantly higher proportion of infected women compared to uninfected women under the age of 35 years was observed, with these 2.29 times more likely to have HPV than those over 35 years of age (OR = 2.29; CI_95%_ = 1.35–3.90). Regarding lesion grades groups (infected women), differences in age range distribution were not observed. However, these women with three or more pregnancies were more likely to have LSIL (OR = 5.64; CI_95%_ = 1.56–20.39) and HSIL (OR = 3.30; CI_95%_ = 1.31–8.28) than those with up to two pregnancies. Furthermore, women who had had less than three lifelong sexual partners were less likely to have HSIL (OR = 0.41; CI_95%_ = 0.17–0.99) compared to those who had had three or more partners.

### 3.2. Distribution of Alleles, Genotypes, and Haplotypes of TGFB1 Genetic Variations and Susceptibility to HPV Infection and Cervical Lesions

Alleles and genotype distribution are given in Table 2. Women were genotyped for *TGFB1* SNVs c.–1638G>A, c.–1347C>T, c.29C>T, and c.74G>C. Only one sample (HPV-infected) could not be genotyped for c.–1347C>T SNV and was excluded from further analyses involving this genetic variation. For each SNV, all groups were tested for Hardy–Weinberg Equilibrium and no deviation from expected genotype frequencies was found (*p* > 0.05). Additionally, higher linkage disequilibrium was observed between c.–1347C>T and c.29C>T (D’ = 0.95, r^2^ = 0.63) (Figure 2).

Differences in allele and genotype distributions between groups were assessed by the χ2 test. Differences were noted regarding the c.–1347C>T (*p* = 0.028) and c.29C>T (*p* = 0.006) allele distribution, and c.29C>T genotype distribution (*p* = 0.023) between the infection groups but not the lesion grades groups. The –1347T and 29C alleles and the 29CC genotype were more frequent in HPV-infected than in uninfected women (40.1%, 50.0%, 24.8% and 32.1%, 39.7%, 17.4%, respectively).

Combinations of investigated variants resulted in eight inferred haplotype structures, two of them possibly recombinant (Figure 3). According to the degree of sequence identity with the *Pan troglodytes TGFB1* gene sequence, the most probable ancestral haplotype (named as **1 PAN*) is formed by c.–1638G, c.–1347C, c.29C, and c.74G alleles (for short, GCCG). Differences between groups were found in the frequencies of **4* (GCTG) when infected and uninfected women were compared (*p* = 0.003) (Table 3). For further analysis, only haplotypes over 5% frequency across the study population were tested; they were **4* (GCTG, 48.3%), **3* (GTCG, 34.2%), **5B* (ACTG, 6.0%), and **2* (GCCC, 5.7%). SNV alleles were represented in haplotype structures according to their position in the *TGFB1* gene, following the order c.–1638G>A, c.–1347C>T, c.29C>T, and c.74G>C.

Binary and multinomial logistic regression adjusted for “age range, age at first sexual intercourse, marital status, and sexual partners during lifetime” or “pregnancies, oral contraceptive usage, marital status, sexual partners during the lifetime, and smoking status” was conducted to assess the influence of SNVs on the susceptibility to HPV infection and development of low and high-grade lesions, respectively (Table 4).

Relevant influences were only observed for c.–1347C>T, c.29C>T, and **4* haplotype regarding HPV infection. Both women carrying c.–1347TT and women with –1347CT or TT were more likely to have HPV than –1347CC ones, with respective odds ratios and confidence intervals (95%) of 2.16 (1.10–4.25) and 1.62 (1.03–2.54). Susceptibility to infection was also greater among women carrying 29CT, 29CC, or 29CT + CC when compared to 29TT, with odds ratios and confidence intervals (95%) of 1.77 (1.06–2.97), 2.31 (1.23–4.34), and 1.92 (1.18–3.12), respectively. Regarding haplotypes, women with two copies (homozygotes) of **4* (GCTG) were less likely to have HPV compared to women with no copy of **4* (OR = 0.39, CI_95%_ = 0.21–0.72). Furthermore, **3* (GTCG) in comparison to **4* (the two more frequent haplotypes) evidenced higher susceptibility to HPV infection in women carrying **3*/**4* or **3*/**3* than **4*/**4* (OR = 2.13, CI_95%_ = 1.13–4.00, and OR = 2.81, CI_95%_ = 1.29–6.10, respectively).

### 3.3. Impact of the TGFB1 Haplotypes on Plasma and Cervical Levels of Protein

After having observed the influence of *TGFB1* genetic variations on susceptibility to HPV infection, their impact on plasma and cervical TGFB1 levels was evaluated.

Initially, plasma and cervical levels were found to be higher in infected patients (4575.19 (IQR 4392.34) pg/mL and 53.17 (IQR 56.46) pg/mg of total protein, respectively) than in uninfected (2964.80 (IQR 3091.45) pg/mL and 32.57 (IQR 54.49) pg/mg of total protein, respectively) (*p* < 0.001 and *p* = 0.008, respectively). However, among the lesion groups, there was a difference only in plasma levels (*p* = 0.007), whose group of women with LSIL presented higher TGFB1 levels than the NL group (6653.45 (IQR 5098.44) and 3689.42 (IQR 3383.84) pg/mL, respectively, *p* = 0.010) (Figure 3). Thus, TGFB1 plasma levels were investigated in the uninfected and HPV-infected women groups according to the haplotype structure inheritance (Table 5).

The difference was observed in the HPV-infected group regarding the **3* (GTCG) haplotype. Comparing patients **3* no carriers, homozygotes (**3*/**3*), and heterozygotes (**3*/Other), heterozygotes presented a lower amount of TGFB1 than those with **3* no copy (3067.13 (IQR 4200.20) pg/mL and 4836.23 (IQR 4313.38) pg/mL, respectively, *p* = 0.03). There was also a lower TGFB1 level in **3* carriers (homozygotes + heterozygotes) when compared with **3* no carriers (3993.99 (4173.05) and 4836.23 (4313.38), respectively, *p* = 0.04). For TGFB1 cervical levels, there was no difference neither in uninfected nor in HPV-infected women concerning the haplotype inheritance (Table 6). The results above were very similar when analyses were conducted after outlier exclusion.

## 4. Discussion

As far as we are aware, this was the first study to address the haplotype structures of these *TGFB1* genetic variations in HPV infection and the development of low- and high-grade cervical lesions. It is also the first study to assess the impact of *TGFB1* haplotypes on cervical and plasma TGFB1 levels according to the disease context.

This case-control study comprised 351 women, of whom 161 were HPV-infected and 190 were not. The presence of HPV in the uterine cervix may lead to the development of the intraepithelial lesions and, therefore, only the infected group was included in the cervical lesions analyzes.

Among the extrinsic factors to HPV infection, younger age, cervical intraepithelial lesions, the high number of pregnancies, and sexual partners were in agreement with other studies [23,24]. Genetic factors have also been associated with HPV infection in the cervical microenvironment, especially variations in the genes of immune system components. This research group has found an association between HPV infection and genetic variation in *FOXP3* [25], *CXCL12* [26], *IL-10* [27], and *TGFB1* [28].

*TGFB1* gene regulation and expression levels are affected by the presence of SNVs in the gene locus [29]. The c.–1638G>A SNV is located in the enhancer region 1. Reduced affinity for the cAMP response element-binding protein (CREB) family in the presence of allele A is associated with lower TGFB1 levels [30].

The c.–1347C>T variation is located in the first negative regulatory region and T allele carriers have almost double plasma levels in comparison to C allele carriers. Furthermore, as reviewed by Cebinelli and colleagues (2016) [14], several in vitro studies using *TGFB1* promoter-luciferase reporter plasmids demonstrated that the T allele increases relative luciferase activity, compared to the C allele. One hypothesis is the loss of negative regulation by the T allele, increasing *TGFB1* transcription. It was also reported that the presence of thymine instead of cytosine at this locus increases the bind of Yin-Yang 1 transcription factor (YY1) and hence transcriptional activity.

The c.29C>T and c.74G>C SNVs are located in the signal peptide sequence and cause amino acid substitutions, proline to leucine, and arginine to proline exchanges at positions 10 and 25, respectively. Modifications in the signal peptide amino acid composition could affect its polarity and result in different rates of protein export [31]. Alleles 29C and 74G have been shown to increase TGFB1 serum concentration [32,33,34].

Concerning the frequency of the investigated variants, the HPV-uninfected group (which was also cervical lesion-free) presented the minor allele frequencies very similar to the Southern European population as reported in the Genome Aggregation Database (gnomAD), using the dataset gnomAD v2.1.1 (Controls) [35,36]. Apart from the high miscegenation observed in the Brazilian population, these data evidence the Southern European ancestry in the current Southern Brazilian cohort, a region that was mainly colonized by the European population [37,38].

Previously in HPV infection, Trugilo and colleagues (2018) [28] evaluated only two genetic variations in *TGFB1*, c.29C>T and c.74G>C. They found a higher frequency of 29CC and 74GC genotypes in the infected patients than in the uninfected, with the 29CC/74GC combined genotypes increasing the infection susceptibility. In the current study, 29CC, CT, and CT+CC were independently associated with greater susceptibility to HPV infection than 29TT carriers were. Similarly, Guan and colleagues (2010) [39] reported that American male and female patients with 29CC genotype were more likely to have HPV16-positive squamous cell carcinoma of the oropharynx than 29TT carriers. They also observed that even statistically nonsignificant 74GG genotype distribution was slightly more frequent in HPV16-positive tumor patients than HPV-negative ones. Diversely, other studies in different Brazilian regions with smaller sample sizes evaluated these two genetic variants in HPV-infected and uninfected patients but found no association [40,41,42].

The –1347T allele was associated with infection, as well as –1347TT and CT+TT were in the adjusted analysis. Differently, Guan and colleagues (2010) [39], in the same study previously mentioned, observed that c.–1347C>T genotypes had a similar distribution between the HPV16-positive tumor patients and HPV16-negative control group. Further, Singh, Jain, and Mittal (2009) [43] found that in Indian cervical cancer patients and no cervical lesion controls, the c.–1347C>T allele frequencies were quite similar.

However, the phenotype could be better explained in a natural context by a set of genetic variations, such as the haplotypes, rather than variations studied in isolation. Here, four *TGFB1* SNVs were analyzed and eight haplotype structures were inferred from the current study population. The frequency of haplotypes in the HPV-uninfected group (control) was similar to that found by Vitiello and colleagues (2018) [44] in their cancer-free control group that was part of a study carried out in the same Brazilian region. This agreement may represent with confidence the distribution of *TGFB1* haplotypes in this population.

Of all the eight haplotype structures, **4* (GCTG) was most frequent among uninfected women, being independently associated with protection against HPV infection, as were the single alleles –1347C and 29T harbored by it. The protective effect remained when **4* was compared with **3* (GTCG). Thus, could any haplotype influence protein levels in such a population?

Firstly, plasma and cervical levels of TGFB1 were measured (Table 5). Increased levels were observed in HPV-infected patients (plasma and cervical secretion) and the LSIL group (plasma). HPV and TGFB1 have been closely related: (1) In epithelial cells containing HPV DNA, E6, and E7 oncoproteins can interact with the specificity protein 1 transcription factor (Sp1) and form the E6-Sp1 and E7-Sp1 complexes which can migrate into the nucleus and induce the *TGFB1* gene expression [45,46]; (2) On the other hand, TGFB1 suppresses the long control region (LCR)-driven transcriptional activity and downregulates at the transcriptional level the early HPV16 expression genes [47,48]. In addition to the infected cell, CD4^+^ T cells (TCD4^+^) of cervical tissue also produce TGFB1. Bonin and colleagues (2019) observed that double labeling (CD4^+^/TGFB) by immunohistochemistry was higher in cervical tissue with a high viral load than in uninfected tissue. At the same time, they saw that double labeling CD25^+^/FOXP3^+^ was also higher in the infected cervix [49]. Evidence suggests that CD4^+^ CD25^+^ FOXP3^+^ T regulatory (Treg) cells may play an important role in an immune-tolerant microenvironment and the failure of HPV elimination. Not only were CD25^+^/FOXP3^+^ Tregs producing TGFB, but also TGFB itself induces the conversion of FOXP3^-^ T cells into FOXP3^+^ T cells [50]. It is well known that TGFB1 has a growth-inhibitory effect both on normal epithelial cells and on cells of the immune system, such as proinflammatory T-cells [51]. Thus, TGFB seems to be compromised with an immunosuppressed microenvironment, which could favor the persistence of HPV infection.

Looking for an answer to the question above, the association between *TGFB1* haplotypes and protein levels was assessed in uninfected and infected women. No differences in the TGFB1 plasma and cervical levels were found in uninfected ones according to the haplotype inheritance. Otherwise, **3*/Other patients and **3* carriers (**3*/**3* + **3*/Other) presented lower TGFB1 plasma levels than patients **3* no carriers (Other/Other) in the infected group. Surprisingly, **3* haplotype harbors the –1638G, –1347T, 29C, and 74G alleles that were associated with higher TGFB1 production compared to their respective alternative alleles [30,32,33,34]. However, a possible explanation is that TGFB1 production could be more strongly affected by another unevaluated variation in linkage disequilibrium, inside or outside the gene; or another possibility is that the presence of the –1347T and/or 29C alleles cause a change in affinity for the E6-Sp1/E7-Sp1 complexes. As discussed earlier, these complexes increase *TGFB1* expression, and thus, if the affinity for them is weakened, TGFB1 production could be reduced.

Although the small number of patients with LSIL and the exclusion of patients with cervical cancer may have limited the sensitivity of the cervical lesion grade analyses in this research, the strengths lie in the analysis of haplotypes and the adjustment for potential confounding factors. To our knowledge, this is the first time that *TGFB1* haplotype was associated with HPV infection: **4*/**4* conferring protection against HPV infection as well as **3*/**4* and **3*/**3* increasing susceptibility to HPV compared to **4*/**4* patients. Moreover, infected carriers of **3* presented lower TGFB1 plasma levels than infected patients with no copy of **3*. Although further studies are warranted to confirm the results, the current study suggests the c.–1638G>A, c.–1347C>T, c.29T>C, and c.74G>C *TGFB1* variants as possible elements that can contribute to the understanding of the mechanisms that involve HPV infection and the pathogenesis of cervical lesions.

## Figures and Tables

**Figure 1 cells-12-00084-f001:**
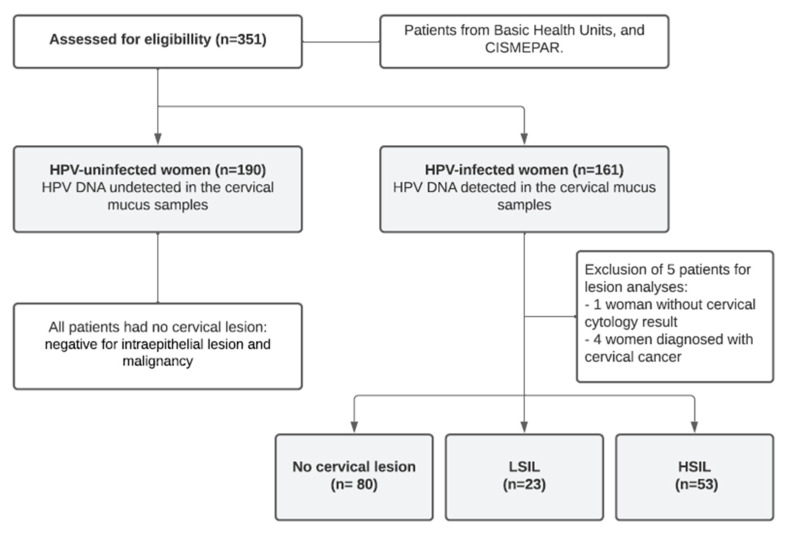
Study flow chart.

**Figure 2 cells-12-00084-f002:**
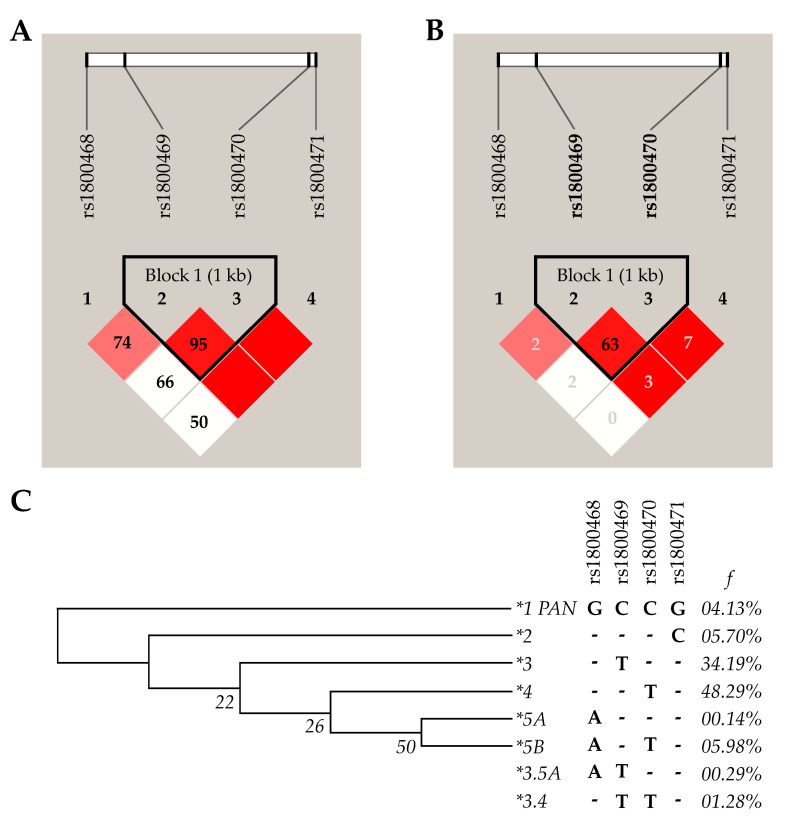
Heatmap linkage disequilibrium and the maximum parsimony analysis of taxa—Values for D’ (**A**) and r^2^ (**B**). The evolutionary history was inferred using the maximum parsimony method (**C**). *TGFB1* SNVs: rs1800468 (c.–1638 G>A), rs1800469 (c.–1347 C>T), rs1800470 (c.29 T>C) and rs1800471 (c.74 G>C).

**Figure 3 cells-12-00084-f003:**
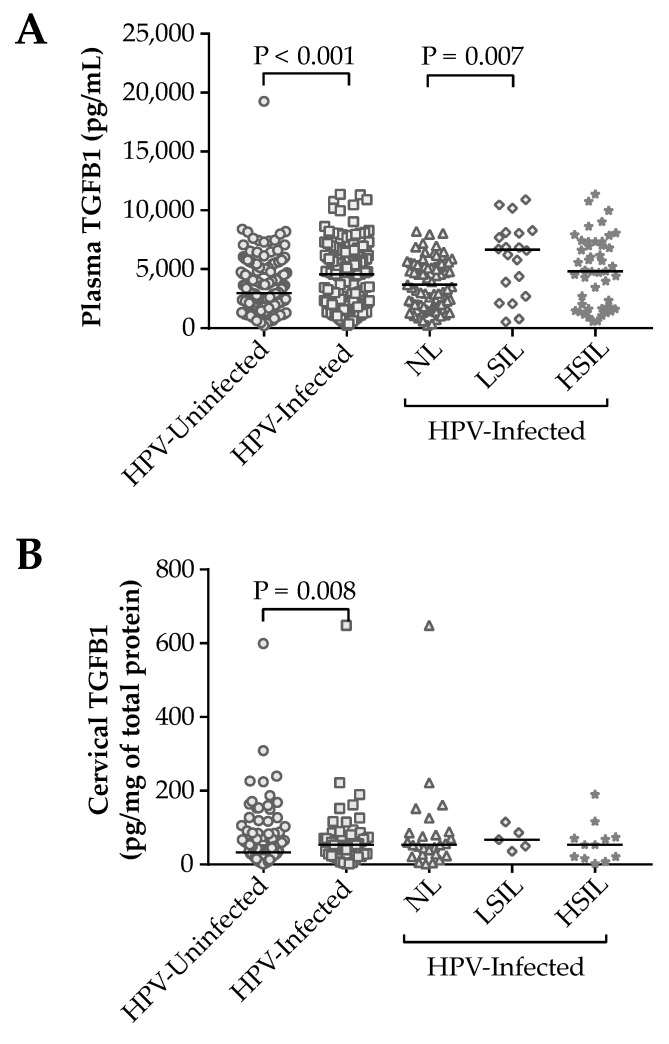
TGFB1 plasma levels according to HPV infection and lesion grade (**A**). TGFB1 levels in cervical mucus according to HPV infection and lesion grade (**B**). Differences between groups were assessed by Mann–Whitney test or Kruskal–Wallis test with Dunn–Bonferroni’s post hoc. *p* < 0.05 considered significant.

**Table 1 cells-12-00084-t001:** Participant characteristics.

Characteristics		HPV			Lesion Grade (Infected Patients)				
		Uninfected (n = 190)	Infected (n = 161)	OR (CI_95%_)	NL (n = 80)	LSIL (n = 23)	HSIL (n = 53)	OR_LSIL_ (CI_95%_)	OR_HSIL_ (CI_95%_)
Age range (years)	<35	57 (30.0)	80 (49.7)	**2.29 (1.35–3.90)**	40 (50.0)	12 (52.2)	26 (49.1)	1.79 (0.51–6.35)	1.16 (0.45–2.95)
	≥35	133 (70.0)	81 (50.3)	Reference	40 (50.0)	11 47.8)	27 (50.9)	Reference	Reference
Age at menarche (years)	<13	86 (45.5)	84 (52.2)	1.16 (0.74–1.83)	42 (52.5)	11 (47.8)	29 (54.7)	Reference	Reference
	≥13	103 (54.5)	77 (47.8)	Reference	38 (47.5)	12 (52.2)	24 (45.3)	1.26 (0.45–3.54)	0.79 (0.37–1.71)
Age at first sexual intercourse (years)	<18	92 (48.4)	99 (61.5)	Reference	47 (58.8)	16 (69.6)	32 (60.4)	Reference	Reference
	≥18	98 (51.6)	62 (38.5)	0.89 (0.54–1.46)	33 (41.3)	7 (30.4)	21 (39.6)	0.87 (0.27–2.74)	1.37 (0.60–3.16)
Pregnancies	<3	108 (56.8)	95 (59.0)	Reference	56 (70.0)	10 (43.5)	28 (52.8)	Reference	Reference
	≥3	82 (43.2)	66 (41.0)	1.27 (0.76–2.12)	24 (30.0)	13 (56.5)	25 (47.2)	**5.64 (1.56–20.39)**	**3.30 (1.31–8.28)**
Oral contraceptive usage	No	132 (69.5)	105 (65.2)	Reference	56 (70.0)	13 (56.5)	32 (60.4)	Reference	Reference
	Yes	58 (30.5)	56 (34.8)	0.96 (0.57–1.62)	24 (30.0)	10 (43.5)	21 (39.6)	3.11 (0.97–9.90)	1.83 (0.78–4.29)
Marital status	Married ^a^	139 (73.2)	97 (60.2)	Reference	48 (60.0)	10 (43.5)	36 (67.9)	Reference	Reference
	Single ^b^	51 (26.8)	64 (39.8)	1.58 (0.98–2.56)	32 (40.0)	13 (56.5)	17 (32.1)	2.47 (0.85–7.15)	0.59 (0.26–1.37)
Sexual partners during the lifetime	<3	111 (58.4)	65 (40.4)	0.63 (0.39–1.00)	37 (46.3)	10 (43.5)	17 (32.1)	1.29 (0.41–4.05)	**0.41 (0.17–0.99)**
	≥3	79 (41.6)	96 (59.6)	Reference	43 (53.8)	13 (56.5)	36 (67.9)	Reference	Reference
Smoking status	No	142 (74.7)	114 (70.8)	Reference	62 (77.5)	13 (56.5)	37 (69.8)	Reference	Reference
	Yes	48 (25.3)	47 (29.2)	1.06 (0.64–1.76)	18 (22.5)	10 (43.5)	16 (30.2)	2.33 (0.74–7.31)	1.12 (0.46–2.76)

Data were analyzed by logistic regression with *p* < 0.05 considered significant (bold). “Uninfected” and “no cervical lesion” groups were reference for HPV infection and lesion grade analysis, respectively. NL, no cervical lesion; LSIL, low-grade squamous intraepithelial lesion; HSIL, high-grade squamous intraepithelial lesion; OR (CI_95%_), odds ratio with 95% of confidence interval. ^a^ Married and civil partner. ^b^ Single, divorced, and widowed.

**Table 2 cells-12-00084-t002:** *TGFB1* genetic variants in HPV infection and cervical lesion status.

*TGFB1* SNVs		HPV			Lesion Grade (Infected Patients)			
		Uninfected (n = 190)	Infected (n = 161)	*p*	NL (n = 80)	LSIL (n = 23)	HSIL (n = 53)	*p*
c.–1638G>A								
	GG	169 (88.9)	140 (87.0)	0.708	68 (85.0)	23 (100.0)	45 (84.9)	0.213
	GA	19 (10.0)	20 (12.4)		12 (15.0)	0	7 (13.2)	
	AA	2 (1.1)	1 (0.6)		0	0	1 (1.9)	
	Allele G	357 (94.0)	300 (93.2)	0.674	148 (92.5)	46 (100.0)	97 (91.5)	0.136
	Allele A	23 (6.0)	22 (6.8)		12 (7.5)	0	9 (8.5)	
c.–1347C>T								
	TT	22 (11.6)	28 (17.4)	0.104	10 (12.7)	6 (26.1)	11 (20.8)	0.398
	TC	78 (41.1)	73 (45.3)		38 (48.1)	7 (30.4)	24 (45.3)	
	CC	90 (47.4)	60 (37.3)		31 (39.2)	10 (43.5)	18 (34.0)	
	Allele T	122 (32.1)	129 (40.1)	**0.028**	58 (36.7)	19 (41.3)	46 (43.4)	0.537
	Allele C	258 (67.9)	193 (59.9)		100 (63.3)	27 (58.7)	60 (56.6)	
c.29C>T								
	CC	33 (17.4)	40 (24.8)	**0.023**	17 (21.3)	5 (21.7)	16 (30.2)	0.768
	CT	85 (44.7)	81 (50.4)		41 (51.2)	13 (56.5)	25 (47.2)	
	TT	72 (37.9)	40 (24.8)		22 (27.5)	5 (21.7)	12 (22.6)	
	Allele C	151 (39.7)	161 (50.0)	**0.006**	75 (46.9)	23 (50.0)	57 (53.8)	0.544
	Allele T	229 (60.3)	161 (50.0)		85 (53.1)	23 (50.0)	49 (46.2)	
c.74G>C								
	GG	173 (91.1)	138 (85.7)	0.117	71 (88.8)	19 (82.6)	45 (84.9)	0.683
	GC	17 (8.9)	23 (14.3)		9 (11.3)	4 (17.4)	8 (15.1)	
	Allele G	363 (95.5)	299 (92.9)	0.128	151 (94.4)	42 (91.3)	98 (92.4)	0.702
	Allele C	17 (4.5)	23 (7.1)		9 (5.6)	4 (8.7)	8 (7.6)	

Data presented as absolute number and percentage. Two-sided χ^2^ test with *p* < 0.05 considered significant (bold). SNV, single nucleotide variant; NL, no cervical lesion; LSIL, low-grade squamous intraepithelial lesion; HSIL, high-grade squamous intraepithelial lesion.

**Table 3 cells-12-00084-t003:** Comparison of the frequency of *TGFB1* haplotypes in HPV and cervical lesion groups.

*TGFB1* Haplotypes	All	HPV			Cervical Lesion Grade (Infected Patients)			
	(n = 702)	Uninfected (n = 380)	Infected (n = 322)	*p*	NL (n = 160)	LSIL (n = 46)	HSIL (n = 106)	*p*
**1 PAN*	0.0413	0.0395	0.0435	0.790	0.0500	0.0435	0.0283	0.685
**2*	0.0570	0.0448	0.0714	0.128	0.0688	0.0870	0.0755	0.914
**3*	0.3419	0.3105	0.3789	0.057	0.3562	0.3695	0.4151	0.620
**4*	0.4829	0.5342	0.4224	**0.003**	0.4375	0.4565	0.3868	0.634
**5A*	0.0014	0.0026	0	1.000	0	0	0	-
**5B*	0.0598	0.0579	0.0621	0.814	0.0750	0	0.0660	0.166
**3.4*	0.0128	0.0105	0.0155	0.739	0.0125	0.0435	0.0094	0.270
**3.5A*	0.0029	0	0.0062	0.210	0	0	0.0189	0.141

Between groups comparison of a haplotype frequency with the sum of the other haplotypes frequency. Two-sided χ^2^ test or Fisher’s exact test when appropriate with *p* < 0.05 considered significant (bold).

**Table 4 cells-12-00084-t004:** Susceptibility for HPV infection, LSIL, and HSIL according to *TGFB1* genetic variations.

TGFB1 SNVs		Adjusted Odds Ratio (OR (CI_95%_))		
		HPV Infected	Lesion Grade (Infected Patients)	
			LSIL	HSIL
c.–1638G>A				
	GA vs. GG	1.13 (0.56–2.26)	-	-
	AA vs. GG	1.03 (0.09–11.91)	-	-
	GA + AA vs. GG	1.12 (0.57–2.19)	-	-
c.–1347C>T				
	TC vs. CC	1.47 (0.91–2.37)	0.44 (0.14–1.45)	0.95 (0.42–2.17)
	TT vs. CC	**2.16 (1.10–4.25) ***	1.50 (0.39–5.78)	1.76 (0.57–5.38)
	TC + TT vs. CC	**1.62 (1.03–2.54) ***	0.66 (0.23–1.88)	1.11 (0.51–2.43)
c.29C>T				
	CT vs. TT	**1.77 (1.06–2.97) ***	1.52 (0.42–5.44)	1.01 (0.41–2.52)
	CC vs. TT	**2.31 (1.23–4.34) ****	1.33 (0.29–6.07)	1.48 (0.52–4.19)
	CT + CC vs. TT	**1.92 (1.18–3.12) ****	1.46 (0.43–4.96)	1.15 (0.49–2.71)
c.74G>C				
	GC vs. GG	1.60 (0.80–3.20)	1.67 (0.44–6.38)	1.02 (0.37–2.85)
*4 (GCTG)				
	Ht vs. no copy	0.93 (0.55–1.56)	0.95 (0.30–3.04)	0.84 (0.37–1.90)
	Hm vs. no copy	**0.39 (0.21–0.72) ****	1.15 (0.27–4.87)	0.66 (0.21–2.06)
	Ht + Hm vs. no copy	0.69 (0.42–1.11)	1.00 (0.33–3.00)	0.79 (0.36–1.74)
*3 (GTCG)				
	Ht vs. no copy	1.48 (0.92–2.38)	0.73 (0.24–2.21)	1.02 (0.46–2.28)
	Hm vs. no copy	1.81 (0.91–3.58) ^§^	1.18 (0.28–5.09)	1.62 (0.53–4.95)
	Ht + Hm vs. no copy	1.56 (1.00–2.43) ^§^	0.83 (0.30–2.30)	1.14 (0.53–2.43)
*5B (ACTG)				
	Ht + Hm vs. no copy	1.12 (0.56–2.23)	-	1.16 (0.40–3.37)
*2 (GCCC)				
	Ht + Hm vs. no copy	1.60 (0.80–3.20)	1.67 (0.44–6.38)	1.02 (0.37–2.85)
*3/*4				
	Ht vs. *4Hm	**2.13 (1.13–4.00) ***	0.83 (0.16–4.19)	1.77 (0.53–5.88)
	*3Hm vs. *4Hm	**2.81 (1.29–6.10) ****	1.34 (0.20–8.66)	2.39 (0.61–9.45)

Logistic regression adjusted for “age range, age at first sexual intercourse, marital status, and sexual partners during lifetime” (HPV infection analysis) or “pregnancies, oral contraceptive usage, marital status, sexual partners during lifetime, and smoking status” (lesion grade analysis), with “uninfected group” or “no cervical lesion group” as reference, respectively. Ht, heterozygote; Hm, homozygote; CI_95%,_ 95% confidence interval. SNVs alleles in haplotype structures follow the order c.–1638G>A, c.-1347C>T, c.29C>T, and c.74G>C. Bolded values are significant with * *p* < 0.05 or ** *p* < 0.01. ^§^ 0.05 < *p* < 0.1.

**Table 5 cells-12-00084-t005:** Plasma level of TGFB1 according to *TGFB1* haplotype models in uninfected and infected women.

*TGFB1* Haplotypes		TGFB1 Plasma Level (pg/mL)					
		n	HPV-Uninfected	*p*	n	HPV-Infected	*p*
**4* (GCTG)							
	Hm	60	3369.30 (3167.17)		28	4762.51 (4959.49)	
	Ht	77	2888.33 (2858.02)	0.55	73	4565.70 (4093.63)	0.72
	No copy	48	2891.69 (3338.91)		46	4352.38 (4697.96)	
	Ht + Hm	137	3004.68 (3076.96)	0.75	101	4576.46 (4289.30)	0.73
	No copy	48	2891.69 (3338.91)		46	4352.38 (4697.96)	
**3* (GTCG)							
	Hm	22	2344.52 (3189.62)		22	4766.67 (4046.06)	
	Ht	72	3084.49 (3667.38)	0.27	63	3067.13 (4200.20) ^A^	**0.03**
	No copy	91	3114.03 (3151.74)		62	4836.23 (4313.38) ^A^	
	Hm + Ht	94	2891.69 (3124.40)	0.45	85	3993.99 (4173.05)	**0.04**
	No copy	91	3114.03 (3151.74)		62	4836.23 (4313.38)	
**5B* (ACTG)							
	Hm + Ht	20	3528.98 (3796.48)	0.17	18	2474.66 (4087.90)	0.27
	No copy	165	2908.18 (2983.04)		129	4706.97 (4414.91)	
**2* (GCCC)							
	Hm + Ht	16	2298.79 (2713.65)	0.45	21	4831.49 (4719.33)	0.45
	No copy	169	2974.81 (3205.66)		126	4548.31 (4189.74)	
**3* vs. **4*							
	**3* Hm	22	2344.52 (3189.62)		22	4766.67 (4046.06)	
	Ht	54	2984.74 (3091.14)	0.22	44	3403.58 (3793.59)	0.23
	**4* Hm	60	3369.30 (3167.17)		28	4762.51 (4959.49)	
**5B* vs. **4*							
	**5B* Hm	2	4838.50 (---)		-	-	
	Ht	7	3023.64 (3337.03)		9	3452.63 (3521.60)	0.39
	**4* Hm	60	3369.30 (3167.17)		28	4762.51 (4959.49)	

Data presented as median and interquartile range (IQR). Mann–Whitney test or Kruskal–Wallis test with Dunn–Bonferroni’s post hoc with *p* < 0.05 considered significant (bold). ^A^ pair whose medians are significantly different. SNVs alleles in haplotype structures follow the order c.–1638G>A, c.–1347C>T, c.29C>T, and c.74G>C.

**Table 6 cells-12-00084-t006:** Cervical levels of TGFB1 according to *TGFB1* haplotype models in uninfected and infected women.

*TGFB1* Haplotypes		TGFB1 Cervical Level (pg/mL)					
		n	HPV-Uninfected	*p*	n	HPV-Infected	*p*
**4* (GCTG)							
	Hm	38	35.28 (50.71)		11	66.67 (112.24)	
	Ht	46	29.02 (75.07)	0.87	20	53.17 (49.93)	0.28
	No copy	27	36.37 (38.99)		17	47.45 (64.76)	
	Ht + Hm	84	31.29 (68.09)	0.88	31	60.33 (49.99)	0.24
	No copy	27	36.37 (38.99)		17	47.45 (64.76)	
**3* (GTCG)							
	Hm	13	29.42 (31.88)		7	47.45 (64.83)	
	Ht	39	29.06 (77.90)	0.68	23	52.85 (42.88)	0.68
	No copy	59	35.34 (51.42)		18	63.50 (69.55)	
	Hm + Ht	52	29.24 (53.16)	0.43	30	52.82 (55.31)	0.38
	No copy	59	35.34 (51.42)		18	63.50 (69.55)	
**5B* (ACTG)							
	Hm + Ht	9	30.00 (56.74)	0.83	5	68.05 (111.38)	0.87
	No copy	102	33.90 (55.08)		43	53.15 (51.66)	
**2* (GCCC)							
	Hm + Ht	12	65.05 (86.80)	0.10	5	52.84 (67.79)	0.95
	No copy	99	29.42 (53.92)		43	53.19 (58.06)	
**3* vs. **4*							
	**3* Hm	13	29.42 (31.88)		7	47.45 (64.83)	
	Ht	30	25.03 (98.54)	0.69	14	53.17 (55.35)	0.45
	*4 Hm	38	35.28 (50.71)		11	66.67 (112.24)	
**5B* vs. **4*							
	**5B* Hm	1	-				
	Ht	3	30.00 (---)		3	68.05 (---)	---
	**4* Hm	38	35.28 (50.71)		11	66.67 (112.24)	

Data presented as median and interquartile range (IQR). Mann–Whitney test or Kruskal–Wallis test with Dunn–Bonferroni’s post hoc with *p* < 0.05 considered significant. SNVs alleles in haplotype structures follow the order c.–1638G>A, c.–1347C>T, c.29C>T and c.74G>C.

## Data Availability

All data generated or analyzed during the current study are included in this published article.

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
