# Peer review of "Haplotype Structures and Protein Levels of TGFB1 in HPV Infection and Cervical Lesion: A Case-Control Study"

_cells, 2022, doi:10.3390/cells12010084_

Round 1
Reviewer 1 Report
Dr. Trugilo and collaborators' s article is well written, effectively presents the haplotype analysis and the consequent blood levels of TGFB1 in HPV positive women.
The only weakness I note in the study is the low number of positive patients enrolled, especially women with LSIL.
But this weakness factor is rightly mentioned in the discussion section by the authors.
Therefore I can only agree with what the authors have presented
Author Response
Dear Reviewer
We thank you very much for reading and for the compliments given to our work. We are aware of the sample limitation, but we believe that it was sufficient to carry out the analyzes and to contribute with relevant information about the role of TGFB in cervical cancer.
Reviewer 2 Report
Dear authors, compliments, ggod work and well done.
I haven't question or dubt with the work.
Author Response
Dear Reviewer
We thank you very much for reading and for the compliments given to our work.
Reviewer 3 Report
In this study, the authors analyzed different genetic variations of TGFB1 and found that some haplotypes correlated with HPV infection. They also found that TGFB1 plasma and cervical levels were higher in the infected patients. This reviewer found this study to be of interest to the scientific community, but there are a few issues that warrant major revisions prior to acceptance of the manuscript.
Issues:
1. The use of the subjective *3, *4 etc. in the abstract is confusing since you would need to read the paper to understand what that means.
2. Not much info on HPV in the introduction.
3. This statement here: "epithelium infection by high-risk human papillomaviruses (HR- 50 HPVs) is necessary for cervical cancer development" is not necessarily true, since there are HPV-negative cases of cervical cancer as well as some cervical cancers caused by low-risk types (remember, low risk does not mean no risk).
4. The RFLP data (gels with bands) is not in the manuscript as a figure or supplementary figure and should be included.
5. The use of "190/54.2%" in that format is very odd. 190 over 54.2%? I know that is not what the authors mean, but that is what they are telling me by using that format.
Author Response
Dear Reviewer
We really appreciated the suggestions and appointments made on our manuscript. We have addressed the issues raised bellow.
- The use of the subjective *3, *4 etc. in the abstract is confusing since you would need to read the paper to understand what that means.
R: We agree and the nomenclature of the haplotypes in the Abstract will be changed to facilitate its understanding. *3 represents the haplotype GTCG and *4 the haplotype GCTG. The text was modified as follow:
“Regarding haplotypes, the most frequent were *4 (GCTG) and *3 (GTCG). Women *4/*4 were less likely to have HPV than those with no *4 copy.
- Not much info on HPV in the introduction.
R: We agree and more information about HPV was included in the Introduction as follows:
“Human papillomaviruses (HPVs) are small, non-enveloped double-stranded DNA viruses with an icosahedral capsid capable of infecting skin and mucosa epithelial cells, belonging to the Papillomaviridae family (Tomasino, 2014). Commonly, HPV is the most common cause of sexually transmitted infections worldwide (Fernandes et al., 2022).
More than 200 HPV types have been identified, which are classified into low-risk (LR) and high-risk (HR) HPVs depending on their oncogenic ability (Bernard et al., 2010; NIH, 2022). Persistent infection with high-risk HPVs is associated with several human carcinomas, and considered the main cause of cervical cancer (Tomasino, 2014; Chan et al., 2019), which is the fourth most commonly diagnosed cancer, as well as the fourth leading cause of cancer death in women (Sung et al., 2021).
HPV-driven cancer is a small probability event because most infections are transient and could be cleared spontaneously by the host immune system (Hu and Ma, 2018).
The cervical epithelium infection by high-risk human papillomaviruses (HR-HPVs) is necessary for cervical cancer development, as well as the local immune response is an important determinant of progression and disease outcome (Bosch et al., 2007). Cytokines play a crucial role in mounting and maintaining immune responses against a host of pathogens, including viral infections and tumors (Hardikar et al., 2015).
- This statement here: "epithelium infection by high-risk human papillomaviruses (HR- 50 HPVs) is necessary for cervical cancer development" is not necessarily true, since there are HPV-negative cases of cervical cancer as well as some cervical cancers caused by low-risk types (remember, low risk does not mean no risk).
R: We thank you and agree with the placement. The sentence will be rephrased as demonstrated below:
“Persistent infection with high-risk HPVs is associated with several human carcinomas, and considered the main cause of cervical cancer”.
- The RFLP data (gels with bands) is not in the manuscript as a figure or supplementary figure and should be included.
R: We thank you and agree with the suggestion. An electrophoretic profile figure will be included as supplementary material.
- The use of "190/54.2%" in that format is very odd. 190 over 54.2%? I know that is not what the authors mean, but that is what they are telling me by using that format.
R: We agree with the appointment, so we decided to keep only the percentage data, since the total number of participants was already presented in the description. The text was modified as follow:
“First, 351 women were included in the study and categorized as HPV-infected (45.9%) and HPV-uninfected (HPV control group - 54.1%). The HPV-infected women were divided into three groups based on cytological abnormalities detected and classified according to the Bethesda System classification as follows: no cervical lesion (NL, lesion control group - 51.3%), LSIL (14.7%), and HSIL (34.0%) groups. For lesion analyses, five patients were excluded, one woman without cervical cytology results and four women diagnosed with cervical cancer (Figure 1).”
Round 2
Reviewer 3 Report
The authors have addressed my comments. Thank you!